# Frequency and Correlates of Mild Cognitive Impairment in Myasthenia Gravis

**DOI:** 10.3390/brainsci13020170

**Published:** 2023-01-19

**Authors:** Salvatore Iacono, Vincenzo Di Stefano, Vanessa Costa, Giuseppe Schirò, Antonino Lupica, Bruna Maggio, Davide Norata, Antonia Pignolo, Filippo Brighina, Roberto Monastero

**Affiliations:** 1Department of Biomedicine, Neuroscience and Advanced Diagnostics (BiND), University of Palermo, 90100 Palermo, Italy; 2Department of Experimental and Clinical Medicine, Marche Polytechnic University, 60121 Ancona, Italy

**Keywords:** myasthenia gravis, cognitive impairment, prevalence, neuropsychological testing, depressive symptoms, sleep disorders

## Abstract

Background: Antibodies against acetylcholine receptors (AChRs) can also target nicotinic AChRs that are present throughout the central nervous system, thus leading to cognitive dysfunctions in patients with myasthenia gravis (MG). However, the presence of cognitive impairment in MG is controversial, and the factors that may influence this risk are almost completely unknown. In this study, the frequency of mild cognitive impairment (MCI) in MG, as well as the clinical, immunological, and behavioral correlates of MCI in MG were evaluated. Methods: A total of 52 patients with MG underwent a comprehensive assessment including motor and functional scales, serological testing, and neuropsychological and behavioral evaluation. Results: The frequency of MCI was 53.8%, and the most impaired cognitive domains were, in order, visuoconstructive/visuospatial skills, memory, and attention. After multivariate analysis, only pyridostigmine use was inversely associated with the presence of MCI, while a trend toward a positive association between MCI and disease severity and arms/legs hyposthenia was found. Correlation analyses showed that daily doses of prednisone and azathioprine significantly correlated with depressive symptomatology, while disease severity significantly correlated with depressive symptomatology and sleep disturbance. Conclusions: The presence of MCI is rather frequent in MG and is characterized by multidomain amnestic impairment. Such preliminary data need further confirmation on larger case series.

## 1. Introduction

Myasthenia gravis (MG) is an autoimmune neuromuscular disease characterized by fluctuating musculoskeletal weakness, commonly caused by antibodies against acetylcholine receptors (AChRs), muscle-specific kinase (MuSK), or anti-low-density lipoprotein receptor-related protein 4 (LRP4) at the neuromuscular junctions [1,2]. Although MG is considered a rare disease, it causes significant disability and morbidity due to the involvement of bulbar muscles during myasthenic crisis, leading to dyspnea, respiratory failure, and intensive care unit admission [3]. The most common MG form is due to antibodies against AChRs, expressed at the neuromuscular junction [4]. However, isoforms of AChR subunits are also expressed in the central nervous system (CNS) [5,6]. These findings support the hypothesis that patients with MG may have some clinical manifestations of CNS involvement. Since nicotinic AChRs in the CNS are mainly expressed in the hippocampus, hypothalamus, midbrain, and cerebral cortex, the cholinergic blockade at these CNS levels might cause cognitive impairment [7,8].

Studies that have assessed the frequency and type of cognitive impairment in patients with MG are still scant and have often reported contradictory results. Some authors have described low cognitive performance on tests evaluating memory, executive functioning, and visuospatial abilities in patients with MG [8,9,10,11]. In contrast, others have not confirmed these findings, suggesting that MG patients’ worse performance on cognitive tasks may depend on muscle fatigability, comorbidities, and visual and motor slowness [12,13].

In addition, the availability of more effective treatments resulting in reduced mortality and improved diagnostic accuracy has led to an increase in the mean age of patients with MG [14]. As the population ages, the prevalence of cognitive disorders also increases [15]. Therefore, even in elderly patients with MG, it is possible to find some degree of cognitive impairment, regardless of the mechanisms that sustain the disease itself. Additionally, apart from the cholinergic hypothesis, cognitive impairment in MG might be due to neuroinflammation as occurs during systemic lupus erythematosus, rheumatoid arthritis, and multiple sclerosis [16,17,18]. Furthermore, patients with MG have a two-fold increased risk of developing depressive disorder and a higher incidence of other autoimmune diseases [19,20,21]. Therefore, it is not surprising that depression, sleep disturbances (e.g., obstructive sleep apnea syndrome), higher comorbidity burden, and older age may also cause cognitive impairment in patients with MG.

The construct of mild cognitive impairment (MCI) as a clinical entity was first introduced by Petersen et al. [22] to differentiate healthy control subjects from those with mild dementia and Alzheimer’s disease (AD). The definition of MCI, originally created as a prodrome of dementia and AD [23], was later applied to other neurodegenerative diseases such as Parkinson’s disease and has also been suggested in patients with amyotrophic lateral sclerosis [24,25]. Thus, MCI represents a high-risk clinical condition for dementia, with relevant prognostic and therapeutic implications [23].

Based on these premises, the aims of the present study are: (1) to evaluate the frequency of MCI in MG patients; (2) to describe the association between demographical, clinical, immunological (AChRs and MuSK serum titers), and behavioral (depression and insomnia) correlates in subjects with MG, and (3) to evaluate the different clinical, cognitive–behavioral, and immunological profiles in subjects with ocular vs. generalized MG.

## 2. Materials and Methods

### 2.1. Study Design and Patient Collection

Patients with newly diagnosed MG and those who made regular follow-up visits to the Neuromuscular Unit of the Azienda Ospedaliera Universitaria “Policlinico Paolo Giaccone” in Palermo, Italy, from November 2021 to November 2022 were included in the present study. Only patients older than 18 years were included in the study. The diagnosis of MG was carried out according to current criteria: decremental U-shaped response at 3 Hz repetitive nerve stimulation and/or increased jitter at single-fiber electromyography, testing for autoantibodies against AChRs, MuSK, and LRP4 in all AChR-negative patients [26,27]. All MG patients were screened for the presence of thymoma with CT or MRI scanning of the mediastinum. The disease severity was assessed by using the Myasthenia Gravis Foundation of America (MGFA) classification, while the quality of life was estimated by using the Myasthenia Gravis Activity Day living (MG-ADL) score [28,29]. Finally, patients were classified as early onset (EO, <50 years) and late onset (LO, ≥50 years) [1].

### 2.2. Neuropsychological Assessment and Mild Cognitive Impairment Definition

The neuropsychological evaluation was carried out in a single session of approximately 1 h by an experienced neuropsychologist who was blinded to the clinical data of patients. Patients underwent a “level I” assessment (i.e., of global cognition) using the Frontal Assessment Battery (FAB), a screening test for executive dysfunction [30].

For “Level II” assessment (i.e., multidimensional cognitive assessment), the battery called “Esame Neuropsicologico Breve, 2° version” (ENB-2) was administered [31]. The ENB-2 was standardized with an Italian sample of healthy people of different ages and levels of schooling and includes the following 14 tasks (the relative cognitive ability assessed by each test is shown in parentheses):The Digit Span test ((DS), verbal short-term memory);The immediate and delayed recall of short prose ((SP), verbal/auditory episodic memory);The Memory Interference Test after 10 and 30 s ((MI), working memory);The Trail-Making test, part A ((TMT-A), selective attention);The Trail-Making test, part B ((TMT-B), divided attention);The Token test ((TT), verbal comprehension);The Phonemic Fluency test ((PF]), lexical access);The Cognitive Estimation test ((CE), executive functioning);The Verbal Abstraction test ((VA), logical reasoning and abstraction);The Superimposed Silhouettes Test ((SS), visuoperceptual ability);The Clock Drawing test ((CDT), visuoconstructional ability);The House Figure copy ((FC), visuospatial and constructional abilities);The Daisy Drawing test ((DD), constructional apraxia);The Praxis test ((PT), ideomotor apraxia).

MCI was classified by adapting Petersen’s criteria for MCI to MG as follows: (1) subjective cognitive complaints reported by the patient or an informant; (2) impaired performance on at least one test in each cognitive domain [32,33]. For each test, details regarding administrative procedures and Italian normative data for score adjustment (based on age, gender, and education) were used [30,31]. Neuropsychological performance was considered as impaired when the subject scored 2 standard deviations (SD) below normality cut-off values (3) preserved general cognitive functions (age- and education-adjusted FAB scores within normal range) [30], (4) normal or minimal impairment of MG-ADL, and (5) an absence of major neurocognitive disorders (DSM-V criteria) [34].

Specifically, the following five cognitive domains were assessed (tests included for each domain are described in parentheses):− Memory (DS, SP, and MI);− Attention (TMT-A and TMT-B);− Language (TT and PF);− Executive functioning (CE and VA); − Visuoconstructive/visuospatial functioning (SS, CDT, FC, and DD).

Although the EBN-2 provides a final total score, this score, as well as the PT score, was not used for the current analyses.

### 2.3. Behavioral Assessment

To assess sleep disturbances and their severity, the Insomnia Severity Index (ISI) was used, a 7-item self-reported questionnaire assessing the nature, severity, and impact of insomnia in the last month, wherein a score above 7 points indicates the presence of significant sleep disturbances [35]. To assess the presence and severity of depression, the Beck Depression Inventory (BDI) was used, a 21-item questionnaire assessing the severity of depressive symptoms in the last week ranging from 0 to 63. A higher score indicates greater depression severity, and the cut-off ≥10 points indicates the presence of depressive symptoms [36].

### 2.4. Statistical Analysis

The Shapiro–Wilk test was performed to check normality for all quantitative variables. Continuous variables with non-normal distribution were described as median and interquartile range (IQR), while other normally distributed variables were reported as mean and standard deviation (SD). Qualitative variables are reported as number and relative percentages. Categorical variables were compared using chi-squared test or Fisher’s exact tests, as appropriate. Continuous variables were compared using Mann–Whitney tests or Student’s *t*-test, as appropriate. To evaluate the possible factors associated with MCI, an unconditional logistic regression analysis was performed for each study variable, considering the presence of MCI as the outcome variable. The odds ratios (ORs) with 95% confidence intervals (CIs) and *p*-values (two-tailed test, a = 0.05) were calculated. Multivariate analysis was performed to evaluate the independent effect of each factor after adjustment for confounding. The multivariate model included all predictors associated with the outcome in the univariate analysis with a threshold of *p* ≤ 0.10. Age at baseline, sex, education, and disease duration were considered as a priori confounders. The model was manually constructed using the likelihood ratio test (LRT) to compare the log-likelihood of the model with and without a specific variable. Correlation analyses between continuous variables were carried out by using Spearman correlation coefficients (rs). Analyses were performed using SPSS (IBM Corp. Released 2019. IBM SPSS Statistics for MacOS, Version 26.0. Armonk, NY: IBM Corp), and the level of significance was set at *p* < 0.05.

## 3. Results

A total of *n* = 52 participants completed the neuropsychological and behavioral assessment (mean age 56.9 ± 15 years; *n* = 30 males, 57.7%).

### 3.1. Frequency of Mild Cognitive Impairment

Twenty-eight of the fifty-two enrolled MG patients were classified as having MCI according to Petersen’s modified criteria, thus reaching a frequency of 53.8% (Figure 1A). The most common impaired cognitive domain was visuospatial/visuoconstructional ability (*n* = 16; 57.2%) followed by memory (*n* = 11; 39.3%), attention (*n* = 10; 35.7%), executive functioning (*n* = 8; 28.6%), and language (*n* = 7; 25%), in order (Figure 1B). Compared with cognitively normal (CN) patients, those with MCI showed a higher frequency of MGFA Class III and IV (*p* = 0.03) and arms/legs hyposthenia (*p* = 0.03), while CN patients had a higher percentage of pyridostigmine use (*p* = 0.03). Positivity for AChRs antibodies was more common in CN than in MCI patients (*p* = 0.01). Table 1 summarizes the differences between CN and MCI patients with MG.

### 3.2. Factors Associated with Mild Cognitive Impairment

After univariate analysis, the presence of MCI was significantly associated with MGFA class III and IV (*p* = 0.04) and arms/legs hyposthenia (*p* = 0.03), while it was negatively associated with pyridostigmine (*p* = 0.04) (Table 2). Multivariate analysis showed a negative and significant association between MCI and pyridostigmine use (OR 0.1; 95% CI 0.01–0.93; *p* = 0.04), while the other positive associations found after univariate analyses did not reach the significance level (Table 2).

### 3.3. Differences between Ocular and Generalized Myasthenia Gravis Patients

A large proportion of patients with generalized MG were taking prednisone (*p* = 0.008) and using intravenous immunoglobulins (IVIg) or plasma exchange (PEX) (*p* = 0.02). Patients with generalized MG showed a higher BDI score (*p* = 0.04). The two clinical phenotypes of MG did not differ regarding the frequency of MCI (45.5% vs. 70.7%; *p* = 0.16) or a specific impaired cognitive domain (Table 3).

### 3.4. Correlation between Myasthenia Gravis Outcomes, Drugs Dosing, and Psychometric Testing

MG-ADL score was found to correlate positively with BDI (rs = 0.32; *p* = 0.02) and ISI (rs = 0.38; *p* = 0.006) but not with FAB (*p* > 0.05). A higher MGFA class was associated with a higher ISI score (rs = 0.28; *p* = 0.047). BDI correlated positively with daily prednisone dosage (rs = 0.4; *p* = 0.01) and inversely with daily azathioprine dosage (rs = −0.57; *p* = 0.007) (Table 4).

## 4. Discussion

The frequency and associated features of MCI in MG patients were evaluated in the present study. The main findings are that (1) MCI is quite frequent in MG patients, affecting more than half of the patients evaluated; (2) the most impaired cognitive domains are, in order, visuoconstructive/visuospatial skills, memory, and attention; (3) pyridostigmine treatment was negatively associated with the presence of MCI, while increased disease severity and the presence of limb hyposthenia were positively associated with MCI, although these latter results were significant only after univariate analysis; (4) comparing the clinical phenotype of MG, patients with generalized disease showed more depressive symptoms than those with ocular MG; however, the two patient phenotypes did not differ in terms of MCI frequency; and (5) overall, disease severity positively correlated with the presence of sleep disturbances, while disease disability positively correlated with depressive symptoms and sleep disturbance. Regarding treatment, a daily dose of prednisone positively correlated with depressive symptomatology, which negatively correlated with a daily dose of azathioprine.

The presence of cognitive impairment is controversial in patients with MG. Indeed, results from recent meta-analyses suggest the presence of cognitive dysfunction in MG [9,37]. However, in a study that used a large MG sample (*n* = 100), the authors reported no significant differences in cognitive performance between patients and controls [13]. Data obtained in animal models have also shown that MG may be associated with cognitive impairment, suggesting that involvement of CNS structures may occur during the disease [38]. However, the precise mechanism underlying cognitive impairment in MG is still unknown. Interestingly, the frequency of MCI according to Petersen’s modified criteria in our cohort was 53.8%, higher than expected for adults aged 60 years or older and even higher than for patients with PD [24,39]. However, such high frequency of MCI in the present study could be attributable to the definition of MCI adopted (i.e., impaired cognitive performance on only one cognitive test). In addition, adopting a multidimensional battery that includes reduced versions of some tests may further account for the high frequency of MCI found. As far as is known, this study is the first that has been purposefully designed to evaluate the frequency of MCI in MG; therefore, comparison with other data is not feasible. However, in a recent Portuguese study, the authors using the Montreal Cognitive Assessment (MoCA)—a first-level screening test for MCI—described a frequency of cognitive impairment of 66.7%, which is rather similar to that described in the present study [10].

Regarding the impairment of individual cognitive domains during MG, in the present study, visuoconstructive/visuospatial skills, verbal short- and long-term memory, and selective and divided attention appeared to be, in order, the most frequently impaired cognitive domains. These data substantially confirm recent data from a Chinese meta-analysis in which the authors described a wide range of cognitive dysfunctions in MG including visuospatial skills, immediate and delayed verbal memory, visual short-term memory, and speed of information processing [9]. In another meta-analysis specifically focused on memory impairment in MG [37], the authors showed that subjects with MG evidenced lower cognitive performance on tests related to immediate and delayed verbal recall ability, and again, these results confirm those described in the present study. More than 25% of patients with MG also showed impaired executive functioning and language. A recent MRI-based study showed a reduction in the grey matter volume in the cingulate gyrus, the inferior parietal lobe, and the fusiform gyrus; these areas are all involved in cognitive functions, especially in memory and executive functions [11]. Although we did not perform brain MRI in all patients evaluated, this evidence could support amnestic and executive dysfunction in MG, explaining the presence of such deficits in our sample.

Concerning specific factors associated with MCI during MG, after univariate analysis, we found that higher disease severity (i.e., MFGA class III and IV) and hyposthenia of the arms and legs were positively associated with the risk of MCI. However, this association was not maintained after multivariate analysis, wherein only a significant negative association was found between MCI and pyridostigmine use. The trend of the association between disease severity and MCI reported in our population is consistent with existing reports describing a positive association between severe MG and cognitive impairment [9]. In support of this hypothesis, a positive correlation was found in the present study between depressive symptomatology and sleep disturbance in patients with MG and disease severity and disability, thus supporting that more severe disease is associated with greater behavioral burden. Overall, the variability in frequency estimates and factors associated with cognitive impairment in MG results from a number of factors including (1) differences in study design (population-based versus hospital-based cohorts and prevalent versus incident cases); (2) the definition of cognitive impairment (based on Level I screening tests of cognition versus use of Level II multidimensional neuropsychological battery; and (3) possible control for confounding in association analyses.

A possible explanation for the presence of cognitive impairment during MG could stem from the central cholinergic effects related to antibodies, with direct damage to areas involved in learning and memory [7]. However, no association between AChRs and MCI or serum titers of AChRs and MuSK with FAB was shown in the present study. Therefore, our results do not support the hypothesis that a blockade of AChRs at the CNS level underlies cognitive impairment in patients with MG as previously hypothesized by the finding of AChRs in the CNS supporting the central cholinergic block in MG patients [5,7,8]. Some authors showed that corticosteroids may induce negative effects on memory and executive functioning [40]. Indeed, in a cross-sectional study conducted by Ayres and colleagues, the authors described an association between worse memory performance, depression, and glucocorticoid use [10]. Although no association was found between cognitive impairment and corticosteroid use in the present study, we found a positive correlation between BDI score and prednisone dosage. In addition, azathioprine was negatively associated with BDI and ISI scores, thus suggesting a beneficial effect of azathioprine as a steroid-sparing agent on behavioral disorders in MG patients. The strongly negative association between pyridostigmine and cognitive impairment is surprising. Although pyridostigmine is a reversible acetylcholinesterase inhibitor with potentially beneficial effects on the CNS, it is unable to cross the blood–brain barrier, unlike rivastigmine, which is used in AD therapy. Therefore, the possibility that pyridostigmine may improve cognitive performance in MG patients will need to be clarified by future randomized clinical trials. Of interest, data from a small clinical trial conducted in AD patients under pyridostigmine therapy (60 mg/4 times daily over 24 h) did not result in significant improvement of cognitive function compared with placebo [41]. Other authors have hypothesized that the reduced physical activity of MG patients, due to muscle fatigue, may result in somatosensory deprivation with consequent stimulation of the sensorimotor system, which is usually involved in various cognitive activities such as abstract word processing and spatial orientation [11,42]. Thus, reduced stimulation of the sensorimotor system could adversely affect synaptic plasticity phenomena and cause some structural changes in the brain, in areas deputed to cognitive functioning [11]. These data may explain the fact that, in the present study, higher disease severity and the presence of arm/leg hyposthenia showed a trend toward association with MCI.

## 5. Strength and Limitations

The major strength of our study lies in the multidimensional assessment including clinical, immunological, cognitive, behavioral, and therapy-related variables. Furthermore, we sought to evaluate the concept of MCI in MG by applying existing criteria for MCI in dementia that have been adapted for MG. For this purpose, we adopted a cut-off score of 2 SD, which has been recommended by previous research as it is associated with a reliable sensitivity and specificity for the identification of MCI [43].

However, several limitations must be considered when interpreting our data. First, the most important limitation of our study is the small sample size, although most previous studies have used smaller MG patient samples than the one assessed in the present study [7,8,10]. Therefore, we could not perform stratified association analyses related to individual cognitive domains vs. putative MG correlates. Similarly, regression analysis could not be performed to test for specific factors associated with ocular or generalized MG. In addition, the small sample size may have reduced the power of some statistical analyses, for example, the trend toward a positive association found between MCI and MG severity. Second, a possible selection bias cannot be excluded due to the hospital-based study design. In particular, the presence of more severe cases attending our center cannot be excluded, and this may possibly have contributed to the high frequency of MCI detected. Third, although analyses were adjusted for major potential confounders, residual confounding (e.g., medical comorbidity, other psychiatric symptoms, etc.) cannot be excluded.

## 6. Conclusions

Cognitive impairment, as well as depressive and sleep disorders, is a relevant clinical aspect in patients with MG and therefore needs attention and investigation. Data from the present study show that MCI is very frequent during MG, with a multidomain amnestic cognitive phenotype. Although the pathogenetic mechanisms underlying cognitive impairment during MG are still unclear, disease severity, MuSK antibody positivity, and arm and leg hyposthenia might be factors associated with an increased risk of MCI in patients with MG, while pyridostigmine therapy seems to be able to reduce this risk. On the other hand, higher daily corticosteroid dosage and greater functional impairment of MG were found to be associated with greater depressive burden and insomnia symptoms. Overall, these results indicate that the common work-up of MG should include a multidimensional cognitive and behavioral assessment. Further longitudinal and population-based studies are needed to define prevalence estimates and risk factors for MCI in MG. Once obtained, such data will have relevant prognostic and therapeutic implications.

## Figures and Tables

**Figure 1 brainsci-13-00170-f001:**
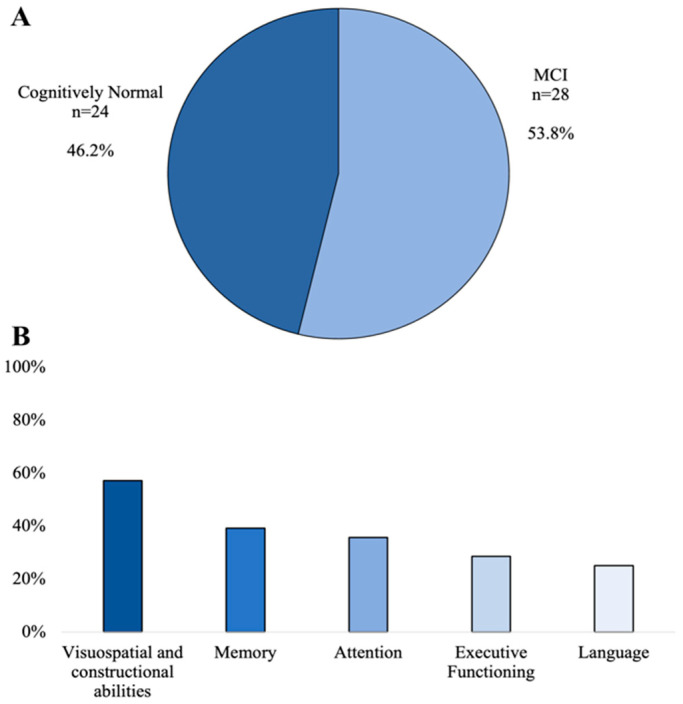
Frequency of MCI (**A**) and impaired cognitive domains (**B**) in our sample.

**Table 1 brainsci-13-00170-t001:** Demographic, clinical, immunological, and cognitive/behavioral differences between CN and MCI patients.

	CN *n* = 24	MCI *n* = 28	*p*
**Demographic and clinical features**			
Male, *n* (%)	14 (58.3)	16 (57.1)	0.93
Age, years, mean ± SD	57 ± 15	56.9 ± 15.4	0.97
Education, years, median [IQR]	13 [8–15]	13 [8–15]	0.37
Disease duration, months, median [IQR]	42 [17–86]	51 [24–138]	0.27
**Clinical features, *n* (%)**			
MG-ADL, median [IQR]	3 [1–6]	3 [1–6]	0.56
Early onset (<50 years)	7 (29.2)	14 (50)	0.13
Generalized MG	18 (75)	23 (82.1)	0.53
AChRs antibody	24 (100)	21 (75)	0.01
AChRs antibody titer, nmol/L, median [IQR]	1.5 [0.97–16]	1.3 [0.75–12.5]	0.39
MuSK antibody	3 (12.5)	8 (26.5)	0.16
AChRs and MuSk antibodies	2 (8.3)	2 (7.1)	1
MGFA Class III-IV	3 (12.5)	11 (39.3)	0.03
Thymic alterations	8 (33.3)	9 (32.1)	0.93
**Prevalent symptoms, *n* (%)**			
Dysphagia	6, (25)	13 (46.4)	0.11
Diplopia	17 (70.8)	18 (64.3)	0.62
Hypophonia	5 (20.8)	10 (35.7)	0.24
Ptosis	13 (54.2)	15 (53.6)	0.97
Dropped Head	2 (8.3)	2 (7.1)	1
Dyspnea	1 (4.2)	3 (10.7)	0.62
Arms/legs hyposthenia	11 (45.8)	21 (75)	0.03
**Treatment**			
Pyridostigmine, *n* (%)	23 (95.8)	20 (71,4)	0.03
Pyridostigmine, mg/daily, median [IQR]	240 [120–360]	195 [120–263]	0.07
Prednisone, *n* (%)	18 (75)	19 (67.9)	0.57
Prednisone, mg/daily, median [IQR]	12.5 [9–19]	12.5 [8–13]	0.36
Azathioprine, *n* (%)	12 (50)	9 (32.1)	0.2
Azathioprine, mg/daily, median [IQR]	100 [50–100]	100 [75–100]	0.7
Mycophenolate, *n* (%)	1 (4.2)	4 (14.3)	0.36
Intravenous Ig or PEX, *n* (%)	12 (50)	15 (53.6)	0.8
**Cognitive and behavioral variables**			
BDI, median [IQR]	10 [4–16]	7 [3–19]	0.94
ISI, median [IQR]	6 [1–9]	4.5 [1–11]	0.66
FAB, mean ± SD	15.6 ± 1.4	15 ± 1.5	0.14

Abbreviations: CN, cognitively normal; MCI, mild cognitive impairment; SD, standard deviation; IQR, interquartile range; MG, myasthenia gravis; MG-ADL, Myasthenia Gravis Activities Daily Living; AChRs, acetylcholine receptors; MuSK, muscle-specific kinase; MGFA, Myasthenia Gravis Foundation America; BDI, Beck Depression Inventory; ISI, Insomnia Severity Index; FAB, Frontal Assessment Battery.

**Table 2 brainsci-13-00170-t002:** Univariate and Multivariate analysis exploring clinical, immunological, and cognitive/behavioral factors associated with MCI in MG patients.

	Univariate Analysis			Multivariate Analysis		
	OR	95% CI	*p*-Value	AdjORs	95% CI	*p*-Value
**Demographic and clinical features**						
Male vs. Female (ref)	0.95	0.3–2.9	0.93	0.4	0.1–2	0.27
Age	1	0.9–1.04	0.97	1.02	0.97–1.1	0.41
Education	0.94	0.82–1.07	0.35	0.95	0.79–1.13	0.54
Disease duration	1	0.99–1.01	0.44	1	0.99–1	0.57
MG-ADL	1.1	0.9–1.28	0.42	-	-	-
Early onset vs. Late onset (ref)	2.4	0.8–7.7	0.13	-	-	-
Generalized vs. ocular MG (ref)	1.5	0.4–5.8	0.53	-	-	-
MuSK antibody vs. AChRs (ref)	7	0.83–60	0.07	2.7	0.4–18.3	0.3
MGFA Class III and IV vs. I–II (ref)	4.5	1.1–18.9	0.04	4	0.8–21.6	0.10
Thymic alterations vs. normal (ref)	0.95	0.3–3	0.93	-	-	-
**Symptoms at MG onset**						
Dysphagia	2.6	0.8–8.5	0.11	-	-	-
Diplopia	0.74	0.2–2.4	0.62	-	-	-
Hypophonia	2.1	0.6–7.4	0.24			
Ptosis	0.98	0.3–2.9	0.97	-	-	-
Arm/leg hyposthenia	3.5	1.1–11.5	0.03	2.7	0.7–11.6	0.17
**Treatment**						
Pyridostigmine use	0.11	0.01–0.9	0.04	0.1	0.01–0.93	0.04
Prednisone use	0.7	0.2–2.4	0.57	-	-	-
Immunosuppressant use	0.64	0.2–1.9	0.42	-	-	-
Intravenous Ig or PEX use	1.16	0.4–3.4	0.8	-	-	-
**Cognitive and behavioral variables**						
FAB score	0.75	0.51–1.1	0.14			
BDI score	1.01	0.96–1.01	0.55	-	-	-
ISI score	1.03	0.94–1.13	0.51	-	-	-
Presence of depression	0.65	0.2–1.9	0.44	-	-	-
Presence of sleep disturbances	0.93	0.3–2.8	0.93	-	-	-

Abbreviations: MCI, mild cognitive impairment; MG, myasthenia gravis; OR, odds ratio; 95% CI, 95% confidence interval; MG-ADL, Myasthenia Gravis Activities Daily Living; MuSK, muscle-specific kinase; AChRs, acetylcholine receptors; MGFA, Myasthenia Gravis Foundation America; FAB, Frontal Assessment Battery; BDI, Beck Depression Inventory; ISI, Insomnia Severity Index.

**Table 3 brainsci-13-00170-t003:** Differences between patients with Ocular and Generalized MG.

	Ocular MG *n* = 11	Generalized MG *n* = 41	*p*-Value
**Demographic and clinical features**			
Male, *n* (%)	8 (72.7)	22 (53.7)	0.32
Age, mean ± SD	62.1 ± 10.4	55.6 ± 15.9	0.25
Disease duration, median [IQR]	44 [16–256]	46 [19–103]	0.87
Early onset, *n* (%)	3 (27.3)	19 (46.3)	0.32
Education, median [IQR]	13 [13–16]	13 [8–15]	0.18
AChRs antibody, *n* (%)	10 (90.9)	35 (85.4)	1
MuSK antibody, *n* (%)	1 (9.1)	10 (24.4)	0.42
MG-ADL, median [IQR]	3 [1–6]	3 [1–6]	0.70
Pyridostigmine, *n* (%)	8 (72.7)	35 (85.4)	0.38
Prednisone, *n* (%)	3 (36.4)	33 (80.5)	0.008
Immunosuppressants, *n* (%)	6 (54.5)	19 (46.3)	0.63
Intravenous Ig or PEX, *n* (%)	2 (18.2)	25 (61)	0.02
**Cognitive and behavioral variables**			
FAB, mean ± SD	15.3 ± 1.6	15.3 ± 1.47	0.78
BDI, median [IQR]	5 [2–9]	11 [5–19]	0.04
ISI, median [IQR]	1 [0–8]	6 [1–11]	0.13
Depression, *n* (%)	2 (18.2)	21 (51.2)	0.09
Sleep disorders, *n* (%)	3 (27.3)	16 (39)	0.73
**MCI frequency, *n* (%)**	5 (45.5)	29 (70.7)	0.16
Memory	1 (9.1)	10 (24.4)	0.42
Executive Functioning	0	8 (19.5)	0.18
Attention	2 (18.2)	8 (19.5)	1
Language	0	7 (17.1)	0.32
Visuospatial and constructional abilities	4 (36.4)	12 (29.3)	0.72

Abbreviations: MG, myasthenia gravis; SD, standard deviation; IQR, interquartile range; AChRs, acetylcholine receptors; MuSK, muscle-specific kinase; MG-ADL, Myasthenia Gravis Activities Daily Living; Ig, immunoglobulin; PEX, plasma exchange; FAB, Frontal Assessment Battery; BDI, Beck Depression Inventory; ISI, Insomnia Severity Index; MCI, Mild Cognitive Impairment.

**Table 4 brainsci-13-00170-t004:** Correlation analyses between cognitive and behavioral testing and demographic, clinical, and immunological variables in MG patients.

	FAB		BDI		ISI	
			*rs*	*p*	*rs*	*p*
Age	−0.2	0.12	−0.14	0.33	−0.05	0.74
Disease duration	0.12	0.39	−0.18	0.2	−0.006	0.97
Education	−0.11	0.43	0.21	0.14	0.17	0.22
MGFA Class	−0.003	0.98	0.18	0.2	0.28	0.047
MG-ADL	−0.01	0.6	0.32	0.02	0.38	0.006
AChRs serum titer	0.22	0.16	−0.26	0.09	−0.14	0.39
MuSK serum titer	−0.24	0.45	−0.3	0.34	−0.19	0.55
Pyridostigmine dose	0.11	0.49	−0.004	0.98	−0.06	0.72
Prednisone dose	−0.06	0.71	0.4	0.01	0.2	0.24
Azathioprine dose	−0.37	0.1	−0.57	0.007	−0.38	0.09

Abbreviations: MG, myasthenia gravis; FAB, Frontal Assessment Battery; BDI, Beck Depression Inventory; ISI, Insomnia Severity Index; MGFA, Myasthenia Gravis Foundation America; MG-ADL, Myasthenia Gravis Activities Daily Living; AChRs, acetylcholine receptors; MuSK, muscle-specific kinase.

## Data Availability

Data are available upon reasonable request to the corresponding author.

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
