# Peer review of "Frequency and Correlates of Mild Cognitive Impairment in Myasthenia Gravis"

_brainsci, 2023, doi:10.3390/brainsci13020170_

Round 1

Reviewer 1 Report

Overall: This is a timely and interesting preliminary study. The results are well presented, the statistical analysis is sound, and the conclusions are sound (and based on the presented data). Although this is a small study, the identification of MCI in such a high percentage of MG patients is interesting to both clinicians and academic researchers. 

Methods: 

·      Enrolment criteria are clear and diagnostic criteria are in line with international guidelines.

·      Statistical analysis is robust, well presented and appropriate

Results: 

·      Table 2: The legend should be expanded to improve transparency and there is a typographical error

·      Table 2: Although 52 participants are included, only 28 have MCI. This impacts the use of univariate and multivariate logistic regression (resulting in several factors approaching significance, and only 1 (pyridostigmine) significant. Although this is not ideal, as this is an initial investigatory study, the data is robust enough for publication and for further expanded studies to be instigated. 

Discussion: 

·      Discussion is well written, based on the presented data and builds a publishable argument. 

Author Response

We thank the Reviewer #1 for his precious comments. All the suggested revisions have been carried out throughout the manuscript and are marked in red.

Reviewer 2 Report

General comments

This manuscript aims at 1. evaluating the frequency of mild cognitive impairment in myasthenia gravis (MG) patients, 2. describing the association between demographical, clinical, immunological and behavioral correlates in subjects with MG and 3. evaluating the different clinical, cognitive-behavioral and immunological profiles in subjects with ocular vs generalized MG. The authors manage to fulfill properly their aims. Of particular value is the fact that the research assesses multidimensionally clinical, immunological, cognitive, behavioral and therapy-related variables.

Minor comments

(line 38 and elsewhere throughout MS, viz., missing space) … admission [3];

(l48) … with MG [8–11].

(l50 and elsewhere throughout MS, i.e., missing full stop) … and motor slowness [12,13].

(l93 and elsewhere throughout MS) Please, do not use acronyms in headings;

(l122-3) … for score adjustment (based on age, gender, and education) were used…

(l127-8) … (DSM-V criteria) [34].

(l141) Please, do not start sentences with acronyms;

(Table 2) Intravenous Ig or PEX… p value… please, add second decimal;

(Table 3) MG-ADL, median [IQR]… p value… please, add second decimal.

Author Response

We thank the Reviewer #2 for his precious comments. All the suggested revisions have been carried out throughout the manuscript and are marked in red.

Reviewer 3 Report

Iacona et al. evaluated the frequency of MCI in MG. Patients with newly diagnosed MG and those who made regular follow-up visits to the Neuromuscular Unit of the Azienda Ospedaliera Universitaria "Policlinico Paolo Giaccone" in Palermo, Italy, from November 2021 to November 2022 were included in this study.

Overall, the study is well designed and the outcomes are appropriate. 

Only a few minor concerns:

Please specify the acronyms in tables, such as CN, FAB in Table 1 and FAB in Table 2.

A few typos: line 99, 20 version; line 128, dis-orders;

Author Response

As noted by Reviewer #3, we enrolled only 52 patients with MG because MG is a rare disease. However, we hope that our preliminary study will be the starting point for prospective studies on large cohorts, even at the population level. Legend in Table 2 has been expanded according to your comment. Thank you.